# The mediating effect of platform width on the size and shape of stone flakes

Sam C. Lin[1,2]*, Zeljko Rezek[3], Aylar Abdolahzadeh[4], David R. Braun[3,5], Tamara Dogandžić[3,6], George M. Leader[4,7], Li Li[8], Shannon P. McPherron[3]

1 Centre for Archaeological Science, School of Earth, Atmospheric and Life Sciences, University of Wollongong, Wollongong, New South Wales, Australia, 2 Australian Research Council Centre of Excellence for Australian Biodiversity and Heritage, University of Wollongong, Wollongong, New South Wales, Australia, 3 Department of Human Evolution, Max Planck Institute for Evolutionary Anthropology, Leipzig, Germany, 4 Department of Anthropology, University of Pennsylvania, Philadelphia, Pennsylvania, United States of America, 5 Department of Anthropology, Center for the Advanced Study of Human Paleobiology, George Washington University, Washington, DC, United States of America, 6 MONREPOS Archaeological Research Centre and Museum for Human Behavioural Evolution, Schloss Monrepos, Neuwied, Germany, 7 Department of Sociology and Anthropology, The College of New Jersey, Ewing, New Jersey, United States of America, 8 Department of Early Prehistory and Quaternary Ecology, Eberhard Karls University of Tübingen, Tübingen, Germany

* samlin@uow.edu.au

**Data Availability Statement:** All relevant data are within the paper and its Supporting Information files.

**Funding:** SCL is supported by the Australian Research Council DECRA Fellowship

## Abstract

To understand the ways in which past stone knappers controlled the morphology of the flakes they produced, archaeologists have focused on examining the effects of striking platform attributes on flake size and shape. Among the variables commonly considered, platform width has routinely been noted to correlate with flake size and hence used to explain past knapping behaviors. Yet, the influence of platform width on flake variation remains equivocal due to the fact that the attribute is not under the direct control of the knapper. Instead, platform width tends to be treated as a by-product of other independent knapping parameters, such as platform depth. In this study, we hypothesize that platform width acts as an intermediary that intervenes the effect of other independent variables on flake attributes. By analyzing experimental flakes produced under both controlled and replicative settings, the results support the hypothesis that platform width mediates the effect of platform depth on flake width, such that flakes with relatively larger platform widths are generally wider but no longer. This finding provides a way to incorporate platform width into discussions of the interrelationships among knapping variables, and highlights the importance of platform width for investigating how past knappers controlled flake production through platform manipulation.

## Introduction

Understanding lithic reduction and its influence on archaeological stone artefact variability is a central imperative of lithic research. To this end, researchers routinely employ replicative flintknapping to reconstruct past reduction procedures and sequences. Today, this approach dominates stone artefact studies. Yet, despite this emphasis on reduction, our knowledge

(DE200100502) (https://www.arc.gov.au/). LL received funding from the European Research Council (ERC) under the European Union's Horizon 2020 research and innovation program (grant agreement n˚ 714658; STONECULT project) (https://erc.europa.eu/). TD received funding from the European Union's Framework Programme for Research and Innovation Horizon 2020 (2014-2020) under the Marie Skłodowska-Curie grant agreement No. 751125 (https://ec.europa.eu/). The funders had no role in study design, data collection and analysis, decision to publish, or preparation of the manuscript.

**Competing interests:** The authors have declared that no competing interests exist.

about the effects of different knapping parameters on lithic geometric attributes is arguably still elementary. While there has been efforts to apply fracture mechanic principles to understand flake formation [1–3], translating the physics of brittle solid fracture to general models that summarize the relationships among empirical flake attributes, particularly those commonly discussed by archaeologists, has arguably not come to pass. A possible reason for this limited application is that fracture mechanic models often involve key variables that cannot be easily extracted from archaeological materials, such as the size and radius of the hammer, impact velocity and the contact time between the hammer and the core [4]. Instead, archaeologists more often study empirical regularities among flakes produced by different experimental conditions to infer the effect of knapping parameters. While this correlative approach does not clarify directly the physical process of fracture underlying flake formation, it helps researchers to gain an understanding of the relationships between the factors under the control of the knapper and the corresponding flaking outcomes.

With respect to this correlative approach, experimental studies using a mechanical flaking setup have made notable strides in demonstrating basic relationships among knapping variables and flake variation [1,3,5,6]. Over the past decade, improvements in the design of these experiments have allowed for high levels of variable control and manipulation [7–9], allowing researchers to investigate the influence on flake variation from a range of factors, including the shape of the platform surface, core surface morphology, the angle and location of percussion, hammer properties, platform beveling, as well as raw material variability and heat treatment [7,10–15]. Among the tested variables, two platform attributes: platform depth (PD) and exterior platform angle (EPA) have repeatedly been shown to exert considerable influence over flake size and shape [7,10,11]. In short, modifying the EPA or PD, or both, during knapping can directly vary the size of the detached flake. When EPA is higher, the resulting flakes tend to be thinner and more elongated than those made with lower EPAs, relative to PD. When EPA is lower, flakes can be detached with higher PD and hence can have greater mass than those made under a higher EPA. This interactive relationship is useful for lithic researchers because the resulting change in flake geometry alters the trade-off economy between flake cutting edge length and reusable flake volume, a property that relates directly to the utility of stone flakes [11,16–19]. Moreover, because the relationship between these two platform variables has been tested repeatedly under various controlled settings, their effect can arguably be applied to examine most, if not all, flaked lithic assemblages across different geographic and temporal settings. Indeed, in a recent study of a large Paleolithic dataset, Rezek and colleagues [20] demonstrated that EPA and PD can effectively track long-term variation in flake edge production over the past two million years across Africa and western Eurasia.

However, in a statistical sense, it is evident that EPA and PD only partially explain the total flake variability [21]. For instance, as shown in Rezek et al. [10], the combined effect of EPA and PD only overrides the influence of core exterior surface morphology up to a certain extent. With respect to platform attributes, the shape of the striking platform has been noted as an important contributing factor of flake variation. Using a geometric morphometric approach, Archer et al. [21] demonstrated that the overall shape of the striking platform can produce relatively more accurate predictions of flake size and shape than with linear measurements like EPA and PD alone. Clarkson and Hiscock [22] examined the relationship between platform area and flake size among flakes with varying platform types. They demonstrated that, although the rate at which flake weight increases with platform size is broadly similar across the different platform types examined, flakes with a focalized platform are consistently smaller than those with a plain platform, while dihedral platform flakes are overall larger. In a recent mechanical experiment, Leader et al. [13] tested the impact of platform beveling in the forms of exterior bevels and lateral bevels (similar to a dihedral shape). The study showed that flake

morphology is significantly impacted by the location of the bevel (exterior vs. lateral) and the depth of the bevel. In general, flakes with a beveled platform tend to have more mass per unit PD than flakes with an unbeveled platform. Moreover, for flakes with concave exterior platform bevels that resemble the so-called 'gull-wing', or 'recessed, U-shaped' platform profile [23,24], the depth of the bevel significantly affects the resulting flake weight—deeper bevel depth leads to greater flake mass relative to PD. These experimental outcomes all suggest that, besides EPA and PD, there are aspects of platform shape that play an important role in controlling flake variation during knapping.

A potential way of capturing the influence of platform morphology in addition to EPA and PD is to include platform width (PW) as an independent variable for summarizing flake variation. Indeed, several studies [25,26] showed that focusing on PW improves the ability for researchers to explain flake variation. Dogandžić et al. [27] demonstrated that when the effects of PD and EPA are controlled in regression models, PW contributes significantly to explaining variation in flake weight, surface area and edge length. In the same study, Dogandžić et al. [27] also reported that flake PW to PD ratio correlates positively with flake thickness (relative to surface area) and negatively with elongation (length/width; due to wider and thinner platform). A similar relationship was described by Dibble [28] among a sample of archaeological flakes, where flakes with relatively wider striking platforms tend to have a greater surface area relative to thickness. However, other studies have failed to observe similar effects of PW on flake attributes. For example, while Dibble [28] observed a positive relationship between PW and mass among a sample of experimental flakes, Shott et al. [29] failed to replicate the same correlation and instead remarked that the effect of PW on flake size is limited in their experimental dataset. Pelcin [30] proposed that PW is a 'threshold variable' that does not directly cause any change in flake size and shape. "...[L]arge flakes may require large platform widths, but a large platform width does not necessarily guarantee a large flake unless all of the other conditions (e.g., platform thickness and exterior platform angle) are met" [28:616]. As such, Pelcin [28:617] argued that PW alone "does not produce large flakes and is thus not a good predictor of flake size."

The ambiguity of PW's effect on flake variation may be related to the fact that PW is not an independent variable like PD and EPA. Rather, PW is a geometric property of the striking platform that is produced after fracture initiation. In other words, PW is not directly under the control of the knapper but rather an outcome of other independent knapping factors such as PD. Indeed, studies have remarked that PW often correlates with PD [7,28]. As a result of this relationship, it can be difficult to determine whether the influence of PW on flake attributes as seen in statistical models is actually a reflection of the effect from PD. It is perhaps for this reason that some studies have employed platform area instead of PD and PW as an explanatory variable for summarizing flake size [29,31–34]. Because both PD and PW are geometric properties of the platform profile, platform area arguably represents a more holistic variable to capture the influence of the striking platform on flake size and shape. However, as noted earlier, experimental studies have shown that the relationship between platform area and flake size changes by platform shape, such that flakes with similar platform areas can have very different mass depending on their platform type [21,22,35]. These findings again suggest that the relationship between platform attributes and flake variation is complicated by additional factors such as the shape of the platform surface.

To clarify these issues, it is necessary to look more closely at the way in which PW forms during fracture. In a recent study, McPherron et al. [36] noted that the fracture propagating from the point of percussion out towards the exterior platform surface occurs at a more or less constant angle, likely stemming from the Hertzian cone angle. They create a new measure, platform surface interior angle (PSIA), to quantify this observation. PSIA values average

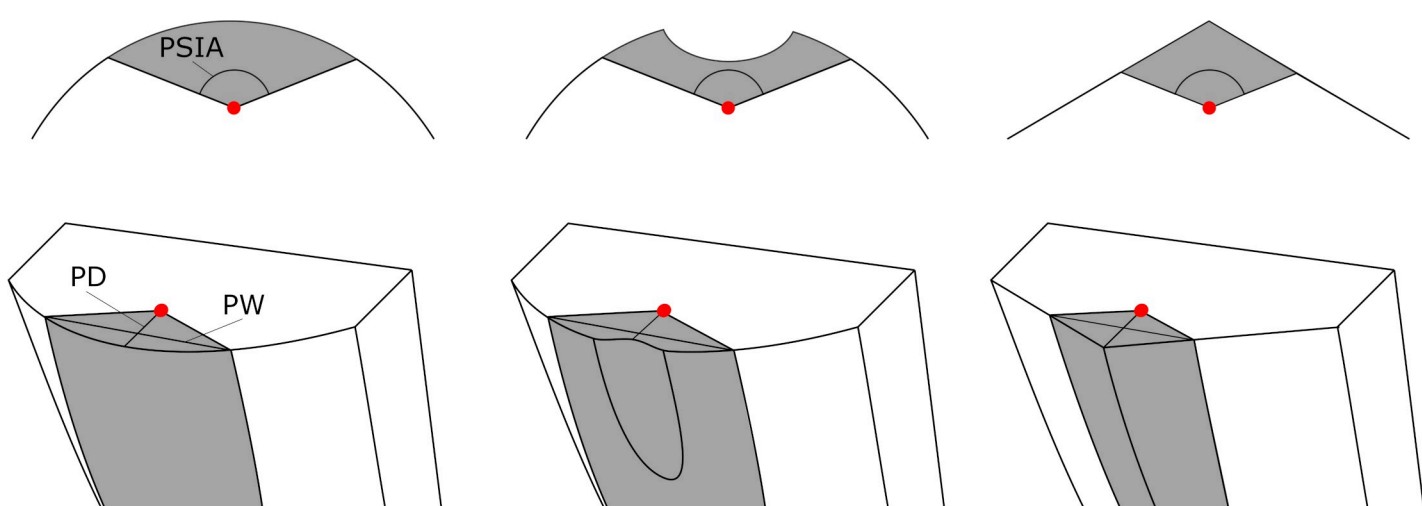

**Fig 1. Schematic views of the relationship among PSIA, PD and PW on three different platform profile shapes: Convex, concave-beveled and triangular/ridged.**
The red dots represent the hypothetical point of percussion, and the shaded areas depict the detached platforms as determined by a PSIA of 136˚.

approximately 136˚ in soda lime glass, which is the same as the Hertzian cone angle in this material [37]. Given that the Hertzian cone angle is a constant, we can model PSIA also as a constant (with some unexplained variability), and this allows us to conceptualize PW as a function of where we strike the core and where the PSIA intersects the exterior platform edge (for a given shape of the platform). As shown in Fig 1, by holding the location of percussion constant, PW can vary notably due to the constant PSIA and the overall platform shape (profile view). This relationship thus may dictate the variation of PW and PD on the detached flake. For example, the concave-beveled platform in Fig 1 has the same PW as the convex platform but with a narrower PD. On the other hand, the triangular platform has the same PD as the convex platform configuration but a much narrower PW.

Based on this view, it is possible that, following hammer impact, the platform surface of a flake is formed after the fracture initiated at the point of impact intersects the edge of the striking platform at a constant PSIA angle. The geometry of the platform, including properties like PD and PW, is thus determined by not only the location of percussion but also the shape of the platform edge. Following the formation of the flake platform, fracture then propagates downward into the core, leading to the detachment of the flake. Undoubtedly, this model is simplistic and does not represent the actual mechanisms underlying the fracture process. However, as a conceptual heuristic, the model can help us hypothesize about the interrelationship between platform variables, such as PD and PW, and flake attributes. For example, because both PW and PD are morphometric attributes of the overall platform geometry, the model predicts that the ratio between PW and PD (hereafter referred to as the PW-PD ratio) should vary by different platform shapes. As illustrated in Fig 1, flakes made on a concave-beveled platform would have a PW-PD ratio that is systematically higher than those made on a convex platform. Likewise, the triangular ridged platform would produce flakes with comparatively narrower platforms and smaller PW-PD ratios. This prediction is consistent with some of the existing experimental and archaeological findings. For example, Leader et al. [13] found in their controlled experiment that, among the concave-beveled flakes, the PW-PD ratio increases as the depth of the platform beveling becomes greater. This is, of course, due to the PD decreasing with this beveling. In the same study, Leader et al. [13] also showed a similar pattern among a

large sample of archaeological flakes, where a concave platform profile tend to have a higher PW-PD ratio than those of a convex and straight profile.

Another prediction we can make concerns the statistical relationship between PW and flake attributes. Even though differences in PW (relative to PD) do not seem to affect the size of flakes in terms of their mass or volume [7,30], they may lead to some variation in flake shape in terms of elongation (flake length-to-width ratio). Looking again at the examples illustrated in Fig 1, we would expect the triangular ridged platform with its narrower PW to produce a flake that is also narrower in width than the flakes associated with the first two platform types. On the other hand, despite having different PW-PD ratios, we would anticipate the flakes made from the convex platform and the concave-beveled platform to share a similar width. In other words, due to differences in platform profile, PW may have a role in influencing the width of a flake. A way to examine this possibility is to consider the interrelationships between PW and flake attributes in the form of a chain, where the effect of one variable on another is mediated or intervened by a third variable. With respect to PW, a possible scenario is that independent factors such as PD, EPA, and platform profile shape, together with the invariable factor of PSIA, first influence the variation of PW. Then, PW in turn influences some aspects of the detached flake, such as flake width. If this is the case, it would suggest that PW is an important variable to consider when explaining flake variation in terms of independent knapping attributes.

In this study, we examine the two predictions derived from our conceptual model of PW formation by using experimentally produced flake assemblages. For the first prediction regarding the PW-PD ratio, we examine a sample of flakes produced in the context of previous controlled studies using cores with different standardized platform profiles [7,10]. With the same experimental dataset, we then evaluate the second prediction about the mediating role of PW by using a mediation analysis to assess whether PW exerts an intermediary effect on the relationship between independent variables, such as PD and EPA, and flake attributes. We also apply the same mediation analysis on a large sample of flintknapped flakes to verify the applicability of the findings to actualistic assemblages. Based on the results, we highlight the importance of platform geometry and discuss implications of the PW-PD relationship in the study of past knapping practices.

## Material and methods

We first examine the two predictions in a sample of glass flakes (n = 150) produced in the context of several previous controlled flaking experiments [7,10,14]. Today, a part of this flake collection is stored at the Max Planck Institute for Evolutionary Anthropology, Leipzig and the University of Wollongong, Australia. The reason for using this dataset is that a number of potentially important variables relevant to our test hypotheses are either controlled or manipulated in the experimental process. These include core exterior morphology, platform profile shapes, exterior platform angle, the angle of blow and the hammer type. Because of its controlled nature, we can be confident that the observed patterns in the dataset are related to the variables in question rather than other confounding factors [9]. In particular, the glass flakes were produced from cores with five different standardized exterior surface configurations: semispherical, convergent, parallel, divergent and center-ridged [10]. These five core forms can be further grouped into three platform types on the basis of the platform profile shape (Fig 2). Based on the predicted relationship between PD and PW outlined earlier, a given PD is expected to produce different PW due to differences in the platform profile alone. If this is the case, we expect flakes produced from the curved and the multi-ridged platforms to produce similar PW-PD ratios because of their similar curvature of the platform profile. Moreover,

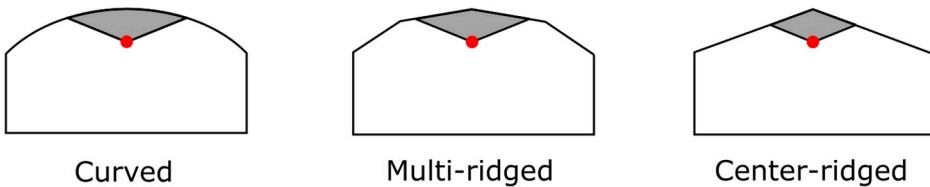

**Fig 2. Schematic views of the three platform profiles (morphologies) represented among the glass flakes examined here.** The red dots represent the hypothetical point of percussion, and the shaded areas depict the detached platforms as determined by a PSIA of 136˚.

flakes made from the curved and multi-ridged platforms should have a PW-PD ratio that is systematically higher than those made from the center-ridged platform.

After assessing the PW-PD ratio among the three platform types, we evaluate whether PW has an intervening effect on the relationships between independent variables, such as PD and EPA, and flake attributes. To this end, we apply a mediation analysis to the glass flake dataset to test for any mediating effect from PW on flake attributes. We acknowledge that this approach does not investigate the actual causal mechanisms underlying the fracture process–such mechanisms will need to be understood from principles of fracture mechanics. Instead, we are interested in characterizing the relationships among the flake variables in question here through statistical modeling. To this end, the mediation analysis enables us to look beyond the simplistic predictor-response relationships by considering more complex interactions among different flake variables. Note that terms such as 'effect', 'influence' and 'impact' are used here strictly to describe statistical relationships among the independent and dependent test variables, rather than the underlying causal mechanisms of fracture.

Developed in the social sciences, a mediation analysis examines the relationships between the independent variable and the dependent variable via the inclusion of a third mediator or intermediary variable [38,39]. In a mediation model, the effect of the independent variable on the dependent is transmitted by the mediator (Fig 3). In other words, the independent variable influences the mediator, which then modifies the dependent variable. In a mediation model, the overall effect of the independent variable (*c* in Fig 3) is separated into two pathways: one leading directly to the dependent variable (the direct effect; *c'* in Fig 3) and another leading to the dependent variable through the mediator (the indirect effect; *a* and *b* in Fig 3). A mediation analysis tests whether the effect of the independent variable on the dependent variable (i.e., the overall effect *c*) is at least partially explained by the chain of effects involving the mediator (i.e., the indirect effects *a* and *b*). A mediating effect implies a temporal sequence, where the

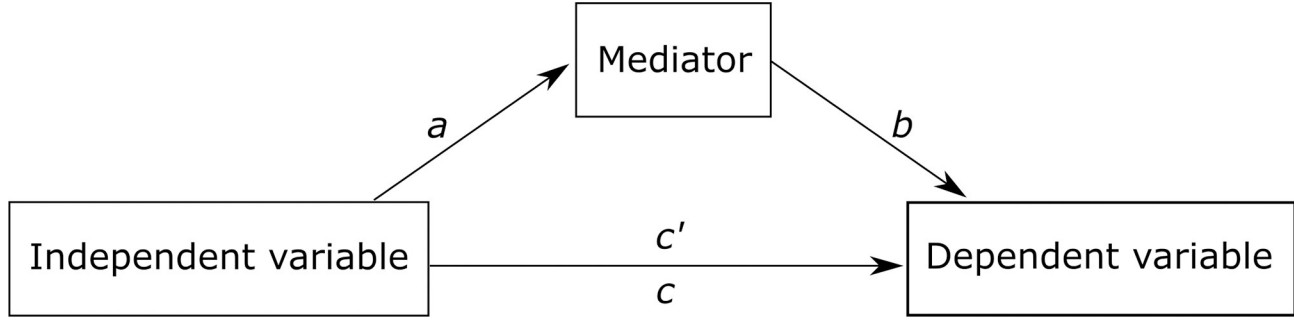

**Fig 3. A mediation model depicting the pathways between an independent variable and a dependent variable.** A mediation analysis tests whether the total effect (*c*) can be at least partially explained by a direct effect (*c'*) and an indirect effect (*a* and *b*) that is mediated through the mediator variable.

independent variable has to influences the mediator first, before the mediator affects the dependent variable [39]. Also, note that a mediation effect is different from an interaction in linear regression analysis. An interaction (also called 'moderation') is concerned with the strength/direction of the effect of an independent variable based on the levels of another independent variable. In our case, for example, moderator variables would be PD, EPA, platform morphology, and highly accentuated core surface morphology, relative to (interacting with) each other. In contrast, a mediation model is focused on clarifying pathway or relationship between the independent and the dependent variables [39].

Since its initial formulation [38], mediation analysis has been applied in a range of science and social science disciplines, from psychology to epidemiology, public health, marketing and education [e.g., 39–42]. Studies have also examined and revised the method, such as clarifying the analytical procedure and introducing more rigorous ways for testing the indirect effect through bootstrapping and other Monte Carlo methods [40,43]. One important concern with mediation analysis has been on the misinterpretation of the test results. Put simply, while a significant mediation test result can indicate the presence of a mediator in the hypothesized model, the same outcome can also relate to other possible pathways [44,45]. Unless all of the alternative models are also tested, empirical confirmation of the hypothesized mediation model cannot be taken as a proof that the underlying hypothesized mediating relationship is true [43,45,46]. Recommendations for the best practice of mediation analysis stress the importance for researchers to explicitly justify mediation hypotheses by explaining why a mediator is needed and which variable should be considered the mediator [40,45,46].

Here, we use the mediation analysis to help explore the statistical relationships among PD, PW and EPA in relation to flake size and shape. As stated earlier, we predict that PW has an intervening influence over the effect of PD and EPA on flake attributes. To examine this possibility, we first apply the mediation analysis to the glass flake assemblage, before repeating the same procedure on a collection of flintknapped flakes. The reason for doing this is to see how well the relationships observed in the mechanically flaked assemblage can be applied to a context of increased variability in flaking conditions akin to that of archaeological materials [9]. The flintknapped assemblage used here was produced in the context of previous studies [47,48], made via hard hammer percussion by multiple knappers of different skill levels. The raw material used was Texas Pedernales River flint as well as flint nodules obtained from the Dordogne region of southwest France. The knapping techniques represented in the assemblage vary from informal freehand reduction to bifacial and discoidal flaking, with no directed end-product goals nor formal core shaping/maintenance. This collection of flakes is currently housed at the University of Wollongong, Australia. A total of 255 complete flakes with a feather termination were selected from this assemblage here for analysis. To increase the size and variability of this flintknapping assemblage, we further included the experimental flakes produced and published by Muller and Clarkson [17]. These flakes were made by both expert and intermediate knappers reducing nodules of Texas chert using a variety of knapping techniques, including Levallois, discoidal, bifacial, bipolar, and blade. For the purpose of this study, we excluded flakes made by bipolar, pressure flaking and punch techniques to avoid potential confounding effects from non-Hertzian fracture. A total of 209 complete flakes with a feather termination made by hard hammer direct percussion were selected from the Muller and Clarkson dataset and added to the flintknapping assemblage, leading to a final sample size of 464.

All analyses were conducted using the R statistical software [49]. For the mediation analysis, we first used linear regression to construct a 'mediator model' to summarize the relationship between the independent predictors (i.e., PD and EPA) and the mediator variable (i.e., PW). Then, a second set of 'outcome models' are constructed to summarize the influence of the

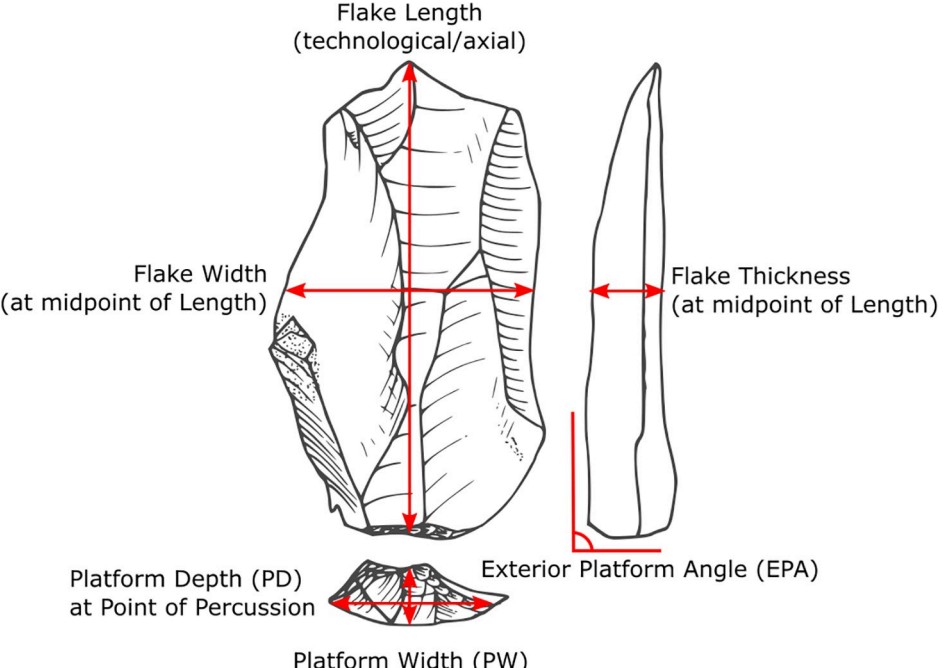

**Fig 4. Flake attributes discussed in the text.**

predictors plus the mediator (i.e., PD, EPA and PW) on the response variable in question. Four response variables are examined in this study: flake weight and the linear dimensions of flake length, width, and thickness (Fig 4). A Gaussian error distribution was used to construct the mediator models. For the outcome models, different distributions were used depending on the response variable examined. For flake weight, either a Poisson or a negative binomial error distribution was used because the data is positively skewed (i.e., many small pieces but fewer large flakes). Because both Poisson and negative binomial models require the response variable to be positive whole numbers, flake weight values are rounded to the nearest gram. For flake dimensional measurements, a Gaussian error distribution was used instead to construct the models.

Before running the linear models, all variables except flake weight were transformed, if necessary, to achieve an approximately symmetrical distribution, and converted to z-scores to allow easier interpretation of the coefficients. All linear models were examined in terms of residual distribution, Cook's distance, leverage, Variance Inflation Factor, and the over/under dispersion in the case of the non-Gaussian models. For the non-Gaussian model, model significance was determined by using a likelihood ratio test [50] and model effect size was measured using the Nagelkerke's pseudo $R^2$, which gauges the degree to which the model parameters improve upon the prediction of a null model containing only the intercept.

After building the mediator models and the outcome models, we used the 'mediate' function from the R package *mediation* [51] to compare the two sets of models to estimate the average causal mediation effects (ACME) of the mediating variable, as well as the average direct effects (ADE) of the predictor variables. The modeled coefficients from the mediation analysis were calculated through bootstrapping over 10,000 iterations. An alpha value of 0.05 was used in this study. However, to account for the inflated Type 1 error due to multiple comparisons, we corrected the critical threshold to 0.003 (over 17 tests) by the Dunn–Šidák correction method. Additional R packages used in the analysis include *car* [52], *ggplot2* [53], *ggpubr* [54],

and *msme* [55]. The R code used for the analyses are included as an rMarkdown file in the supplemental material, along with the data files needed to replicate the results (S1 File). No permits were required for the study, which complied with all relevant regulations.

## Results

### The PW-PD ratio by platform profile

Fig 5 plots the distribution of the PW-PD ratios by platform profile type among the glass flakes. As predicted, the curved platforms and the multi-ridged platforms produced similar PW-PD ratios, while the center-ridged cores produced notably narrower platforms relative to PD. This observation is supported by an analysis of variance (ANOVA) that shows notable differences in the PW-PD ratios among the three platform types ($F(2:147) = 99.41$, $p<0.001$; PW-PD ratio transformed by square-root to achieve a symmetrical distribution to fulfil the assumption of an ANOVA test). A post hoc Tukey test shows that the PW-PD ratios among the center-ridged platforms are lower than those from the other two platform types at $p<0.001$. In contrast, there is no distinguishable variation between the curved and the multi-ridged platforms in terms of their PW-PD ratios ($p = 0.25$).

### The mediating effect of PW on flake attributes

The mediator and outcome models used to carry out the mediation analyses are provided in S1 Table for the glass flakes and S2 Table for the flintknapped flakes. The inclusion of influential cases identified by Cook's distance and leverage in the linear models did not alter the general findings of the mediation tests. As such, the results presented here are based on the entire experimental datasets examined.

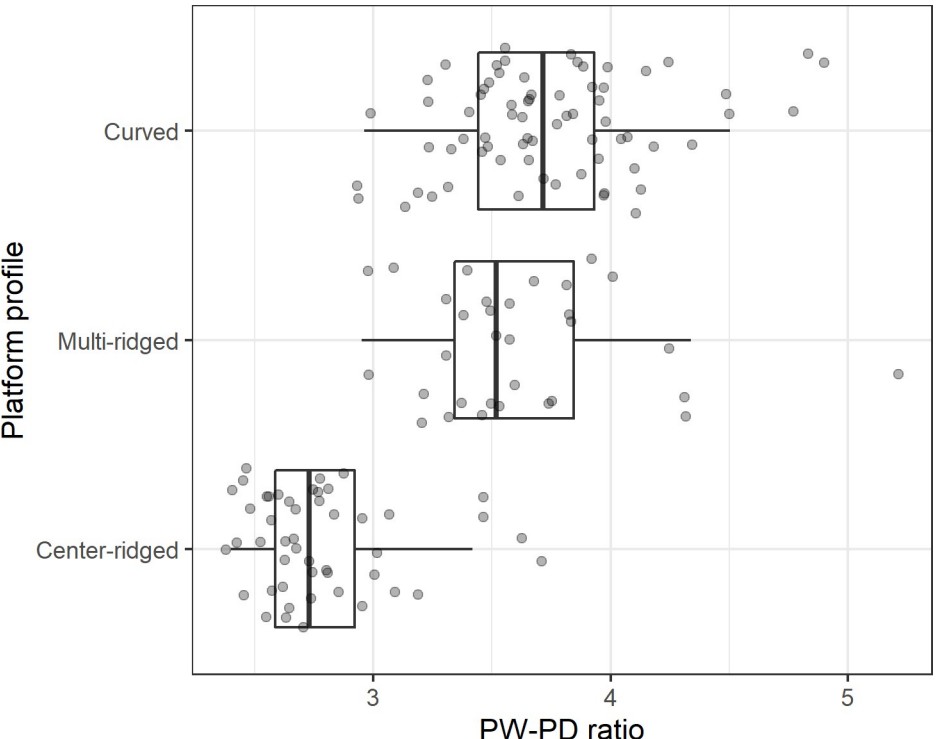

**Fig 5. Boxplot summarizing the distribution of the PW-PD ratio among the glass flakes by platform profile type.**

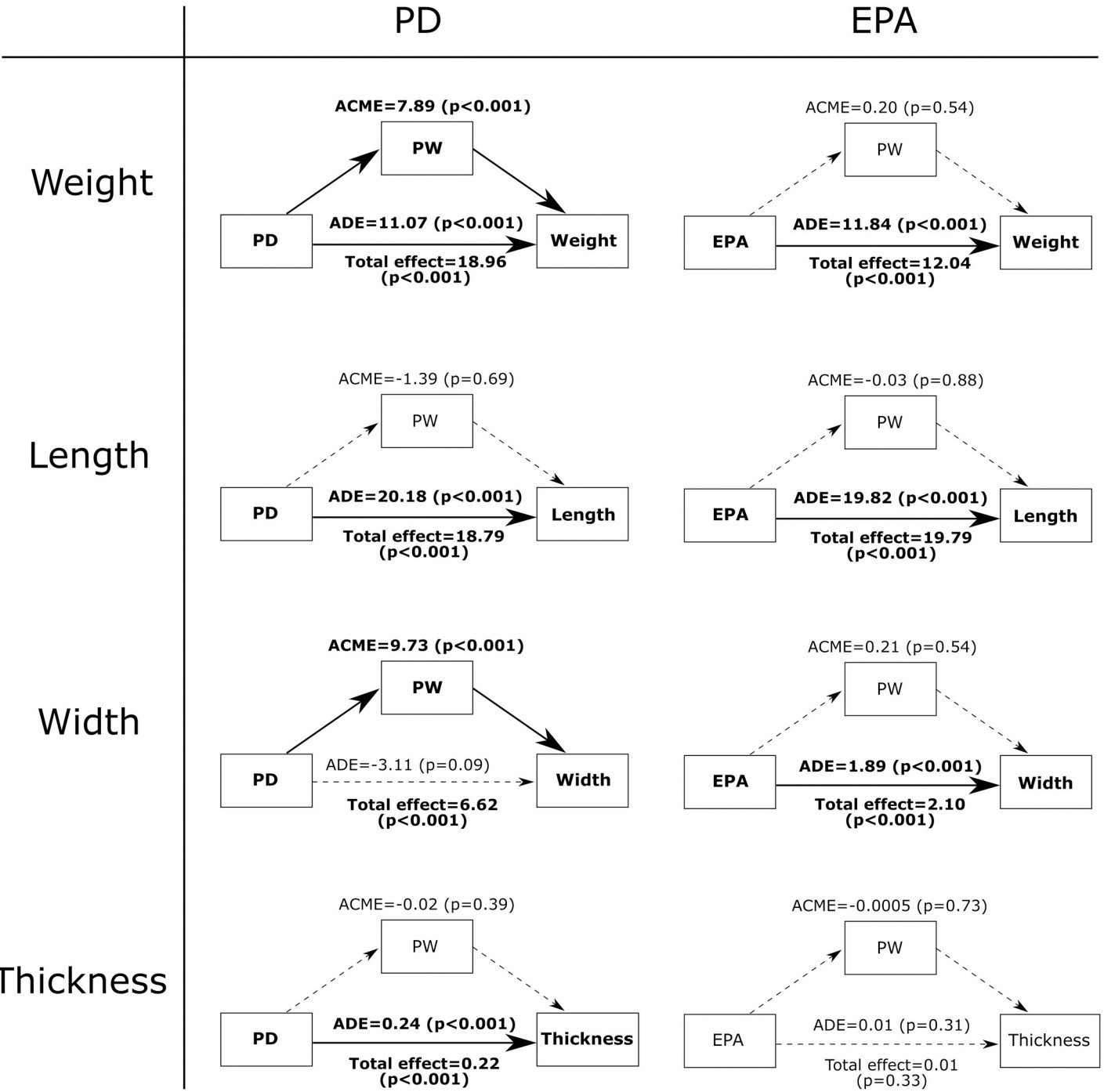

**Fig 6. The mediating effect of PW on the influence of PD and EPA on flake attributes among the glass flake assemblage.** Variable names and arrowed lines in bold indicate pathways that have a p-value below the corrected critical threshold. Dashed lines indicate pathways that have an effect above the critical threshold.

**Glass flakes.** Fig 6 summarizes the mediation analysis on PW with respect to PD and EPA among the glass flakes. Looking first at the results relative to PD, the independent variable has a positive total effect on all four flake attributed examined here. However, for flake weight, 42% of the overall influence from PD first affected PW, which then in turn affected flake

weight. This indirect effect of PW on flake weight likely reflects the dominant influence of PW on flake width. Indeed, the direct pathway of influence from PD to flake width is not significant, whereas the indirect pathway via PW is. This means that the overall positive effect of PD on flake width occurs entirely through PW. In simple terms, PD first influences PW, and then PW in turn determines flake width. In contrast, PW does not have a mediating influence over the effect of PD on flake length and thickness. Likewise, while EPA has a direct positive effect on all of the flake attributes except thickness, none of the effects is mediated by PW.

**Flintknapped flakes.** Fig 7 summarizes the mediation analysis results on the flintknapped flakes. Note that the modeled effect sizes here cannot be compared directly with the results from the glass flakes due to the different transformations used to adjust the variables for linear modelling. The exception to this is flake weight as the variable was left untransformed due to the use of non-Gaussian distributions. The mediation results from here are largely consistent with that of the glass assemblage. Looking at PD, the variable has a positive total effect on all four flake attributes. As with the glass flakes, the relationship between PD and flake weight can be partially (24%) explained by an indirect effect via PW, which again is a reflection of PW's intervening influence on flake width. Specifically, for flake width, 60% of the overall influence from PD first affects PW, which then in turn affects flake width. Although this mediating effect of PW is weaker than the mediation seen earlier among the glass assemblage, the indirect effect still constitutes the majority of the overall relationship between PD and flake width.

Differing to the glass assemblage, a mediating effect of PW was also detected with respect to the relationship of PD and flake thickness among the flintknapped flakes, though the indirect influence constitutes a relatively small proportion (15%) of the overall effect. On the other hand, PW plays no notable role in the influence of EPA on the flake attributes. Unlike with the glass flakes, however, EPA does have a notable impact on the thickness of the flintknapped flakes.

## Discussion

In this study, we used experimental datasets to evaluate two predictions concerning the relationship of PW with respect to other independent knapping variables. The first being that the configuration of PW is related not only to PD but also the profile shape of the striking platform. We used the concept of PSIA [36] and predicted that flakes made on a triangular ridged platform should exhibit systematically narrower PW (relative to PD) than those made on platforms that are more circular in profile. The results from the glass flakes match our predictions. Taking together similar observations by Leader et al. [13] regarding beveled platforms, we conclude that our hypothesized relationship between PW, PD and PSIA is a useful model for conceptualizing the formation of PW during flake production. According to this model, while PW is expected to correlate with PD, it should also exhibit variation depending on platform profile shape.

The second prediction tested is that PW has an intervening influence on the statistical effects of PD and EPA on flake attributes. The results of the mediation analysis support this prediction for PD but not EPA. Specifically, we found that PW has a detectable impact on the effect of PD with respect to flake width and flake weight. Put simply, when PW is relatively wider, the overall effect of PD on flake mass and flake width is amplified, leading to the production of larger and wider flakes. In some respect, this finding is not entirely surprising as previous studies have already noted the explanatory power of PW in regression models to account for flake variation. However, a main obstacle faced by previous studies is the difficultly of relating PW to other independent knapping variables. Specifically, while we can quite easily understand how independent variables like PD and EPA can have a direct influence on flake

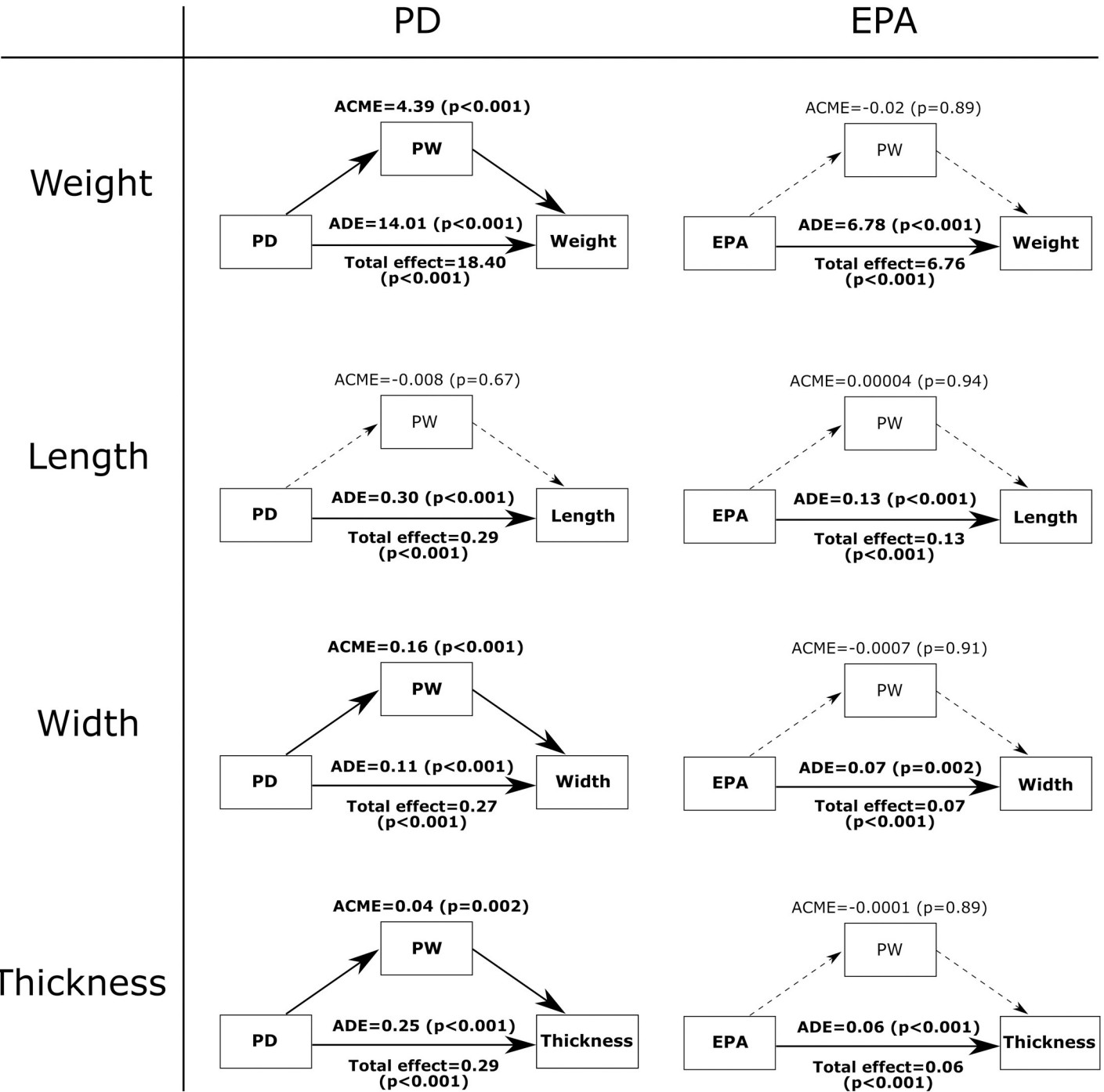

**Fig 7. The mediating effect of PW on the influence of PD and EPA on flake length, width, and thickness in the flintknapped assemblage.** Variable names and arrowed lines in bold indicate pathways that have a p-value below the corrected critical threshold. Dashed lines indicate pathways that have a p-value above the critical threshold.

attributes, it is harder to make sense of the role of PW as the variable is not directly under the control of the knapper. The mediation analysis here helps clarify this question by showing the possibility that PW mediates the influence of PD on flake attributes, namely flake width and flake mass. In these cases, a certain portion of the overall effect of PD goes to influence PW,

and PW in turn helps determine the resulting flake width and flake weight. The fact that this outcome is consistent between the mechanically produced glass flakes and the flintknapped flakes means the mediating effect of PW with respect to flake width and flake weight is likely fundamental and has high external validity [9]. This finding also explains why earlier studies recorded equivocal outcomes when attempting to identify the effect of PW on flake attributes by controlling for PD. If the effect of PW is derived from independent variables like PD, then controlling for PD would inevitably obscure the influence of PW.

It is important to note that the hypothesized mediation relationship in this study only concerns the variables examined here, and the formation of the platform surface geometry and the overall flake morphology undoubtedly involves a multitude of other factors that have not been considered. Moreover, while our statistical results have no immediate bearing on clarifying the actual mechanics of flake fracture, we argue that the results of the mediation analysis can help researchers heuristically conceptualize the influence of knapping variables on flake attributes. As outlined earlier, we propose that the geometry of the flake striking platform is determined by a combination of the location of percussion, the PSIA specific to the brittle solid, and the shape of the platform edge. Following the formation of the flake platform, fracture propagates downward into the core. The fracture is often described to travel along a cleavage plane at an angle controlled by the force angle and the angle of hammer blow [1,2,56]. Given a certain fracture plane angle, the length of the resulting flake should be determined largely by the location of percussion (i.e., how far back from the platform edge did the hammer strike) and the exterior platform angle (i.e., steeper platform edge angle allows fracture to travel farther before existing the core). This model is consistent with our results here showing that the effect of PD and EPA on flake length is not mediated via PW. On the other hand, the strong mediating effect of PW detected in relation to the relationship between PD and flake width supports a scenario that the width of a flake is at least partially controlled by the width of the flake platform. It is currently unclear why PW also mediates the effect of PD on flake thickness among the flintknapped flakes. A possible explanation is that some of the factors in the flintknapped assemblage that influence PW also have an impact on flake thickness. For example, it has been suggested that flakes with a concave-beveled platform also have a more pronounced bulb of percussion [23]. As such, the flintnapped assemblage could contain flakes with a concave-beveled platform that are not only wider due to the larger PW relative to PD, but also thicker due to the greater bulb size.

Our findings here concerning the role of PW and its relationship with other flake attributes have several implications. First, the changing relationship between PW and PD among different platform shapes means that the PW-PD ratio can be used as a proxy for gauging platform variation among archaeological flakes. Second, the positive mediating effect of PW on flake width means that a relatively wider PW would lead to greater flake width and hence surface area. As such, these flakes would also be relatively thinner than are those with a narrower PW. This outcome is consistent with the observation by Dogandžić et al. [27] mentioned earlier, where flakes with higher PW-PD ratios tend to have higher blank area to thickness ratio but lower elongation ratio. The same relationships are observed in the flintknapped assemblage examined here, where the PW-PD ratio has a positive correlation with blank surface area to thickness ratio (Pearson's correlation: $r = .20$, $p<0.001$; blank surface area transformed by square root to standardize the dimension with respect to thickness) and a negative correlation with flake elongation (i.e., length to width ratio) (Pearson's correlation: $r = -0.24$, $p<0.001$). Because these changes in flake morphology implicate the amount of useable edge and mass on a given flake, the PW-PD ratio could be a useful parameter to use for investigating past knapping practices in relation to the management of lithic utility, particularly with respect to the manipulation of platform geometry during flake production [11,17,20]. For instance, we may

expect reduction strategies geared towards making large, broad flakes, such as the Levallois technology, to generate flake assemblages with relatively high PW-PD ratio, while reduction strategies that utilizes parallel ridges for blade production are expected to be associated with relatively low PW-PD ratios.

The results here also show that research efforts toward explaining flake variability with platform attributes has to account for the shape of the flake platform [21,22], as the platform geometry alters the relationship between flake attributes and platform variables such as PW and PD. To this end, conventional platform categories (e.g., plain, dihedral, punctiform, etc.) may not be sufficient to capture the necessary information about the overall platform shape, as these classifications often confound different aspects of the platform geometry [13]. For example, while plain and dihedral platforms describe the topography of the platform surface, they do not say anything about the plan view profile of the platform. Similarly, while facetted platforms have traces of platform preparation, the actual shape of the platform can vary widely. Given the importance of platform profile suggested here, it would be useful to record the plan view profile of striking platforms (e.g., concave, convex, straight, ridged) separately from the platform surface morphology (e.g., plain, dihedral, chapeau de gendarme, etc.) and modification (e.g., facetted). Using existing data collected on a large sample of archaeological flakes from the Middle Paleolithic site of Roc de Marsal (Dordogne, France) [47,57,58], Fig 8 plots the distribution of the flake PW-PD ratio, the elongation ratio and the flake surface area to thickness ratio by a general classification of platform profile. Matching our predictions, the flakes with a concave platform profile tend to have a higher PW-PD ratio and a higher flake surface area to thickness ratio, while those with a ridged platform profile have the lowest PW-PD ratio but the highest elongation ratio. These results not only demonstrate that the PW formation model proposed here can be extended to explain archaeological data, but more importantly, that documenting flake platform profile, even with a simple classification system like the one shown here, can be useful to track fundamental patterns of lithic variation.

## Conclusion

Recent mechanical experiments have made substantial progress in unravelling the relationships between independent knapping variables and flaking outcomes [7,10–14]. Here, we further demonstrate the value of combining this controlled experimental approach with data generated under an actualistic flintknapping setting to address unresolved questions in lithic reduction, in this case the role of PW in flake formation. The consistent analytical outcomes across the two datasets suggest the finding here on PW represents a basic aspect of the flake production process. The intermediary effect of PW on flake width and weight helps explain previous uncertainties regarding the relationship between PW and other knapping attributes. Finally, because the indirect effect of PW implicates the proportion of useable edge on a given flake by influencing the ratio of width to thickness, the PW-PD ratio may be a useful measure for investigating how past knappers managed lithic utility through platform manipulation during flake production.

The mediation analysis employed in this study offers an alternative approach for researchers interested in investigating the connection of multiple variables. Unlike conventional regression that focuses on quantifying the overall effect of independent predictors on singular outcomes, a mediation analysis evaluates the interrelationship among the variables in a more complex form by including indirect effects. However, it is also important to bear in mind the need to explore alternative models involving other potential mediators [43,45]. In this sense, the PW mediation model examined here captures but one aspect of the overall complexity, and there are undoubtedly other potential mediating relationships involved. To this end, a

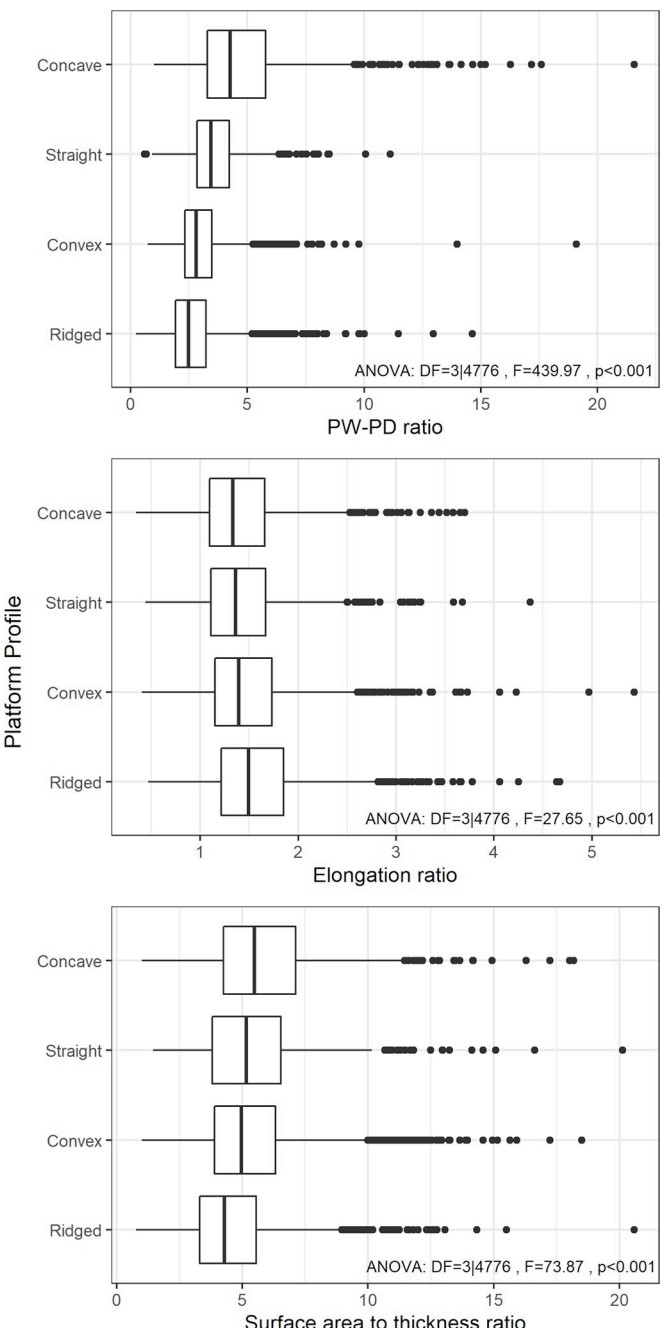

**Fig 8. The distribution of the PW-PD ratio, the elongation ratio and the flake surface area to thickness ratio by platform profile type among the flake assemblage from Roc de Marsal (Dordogne, France).** Only complete flakes are included in the analysis. For the analysis of variance (ANOVA), all three variables were transformed to achieve an approximately symmetrical distribution to meet the assumption of the test.

possible candidate for another mediator to be investigated could be the bulb of percussion. While bulb characteristics are commonly discussed in relation to the effect of hammer material [12,59,60], recent mechanical experiments have shown that, among Hertzian-fractured flakes, bulb size varies by EPA and is correlated with changes in flake properties such as elongation and thickness [11]. From a mediation analysis perspective, the formation of the bulb of

percussion may represent an intervening factor that modifies the effect of knapping variables on flake variation. A mediation analysis would help clarify this relationship.

## Supporting information

**S1 Table. Summary of the mediator and outcome linear models used to carry out the mediation analysis on the glass flakes.**
(PDF)

**S2 Table. Summary of the mediator and outcome linear models used to carry out the mediation analysis on the flintknapped flakes.**
(PDF)

**S1 File. Compressed file containing the rMarkdown file and the data files required for reproducing the statistics and figures in this study.**
(ZIP)

## Acknowledgments

This paper is an outcome of a conference presentation at the session entitled "The Role of Experiments in Lithic Technology" in the 2017 International Symposium of Knappable Material (Buenos Aires, Argentina). We thank the session organizer Daniel Amick for the invitation. Harold Dibble was an author on the original conference presentation on which this paper is based. Sadly, he unexpectedly passed away in June 2018. We would like to dedicate this work to Harold, whose pioneer research in flake formation and lithic experimentation paved the way for the research presented here. Two anonymous reviewers provided valuable feedback that helped improved the clarity of the paper. SCL thank Dr. Sonia Ip for introducing the concept of mediation analysis.

## Author Contributions

**Conceptualization:** Sam C. Lin.

**Data curation:** Sam C. Lin, Zeljko Rezek, Aylar Abdolahzadeh, Tamara Dogandžić, George M. Leader, Li Li, Shannon P. McPherron.

**Formal analysis:** Sam C. Lin, Shannon P. McPherron.

**Methodology:** Sam C. Lin, Zeljko Rezek, Shannon P. McPherron.

**Visualization:** Sam C. Lin, Shannon P. McPherron.

**Writing – original draft:** Sam C. Lin, Zeljko Rezek, Shannon P. McPherron.

**Writing – review & editing:** Sam C. Lin, Zeljko Rezek, Aylar Abdolahzadeh, David R. Braun, Tamara Dogandžić, George M. Leader, Li Li, Shannon P. McPherron.

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
