## [Decision Letter · Decision Letter 0]

26 Jul 2021

PONE-D-21-21600

The mediating effect of platform width on the size and shape of stone flakes

PLOS ONE

Dear Dr. Lin,

Thank you for submitting your manuscript to PLOS ONE. After careful consideration, we feel that it has merit but does not fully meet PLOS ONE’s publication criteria as it currently stands. Therefore, we invite you to submit a revised version of the manuscript that addresses the points raised during the review process.

All comments need to be addressed before re-submission.

We look forward to receiving your revised manuscript.

Kind regards,

Peter F. Biehl, PhD

Academic Editor

PLOS ONE

Journal Requirements:

2. In your manuscript, please provide additional information regarding the specimens used in your study. Ensure that you have reported specimen numbers and complete repository information, including museum name and geographic location.

For more information on PLOS ONE's requirements for paleontology and archaeology research, see https://journals.plos.org/plosone/s/submission-guidelines#loc-paleontology-and-archaeology-research.

Additional Editor Comments (if provided):

All comments need to be addressed before re-submission.

Reviewers' comments:

Reviewer's Responses to Questions

**Comments to the Author**

1. Is the manuscript technically sound, and do the data support the conclusions?

Reviewer #1: Partly

Reviewer #2: Yes

2. Has the statistical analysis been performed appropriately and rigorously? 

Reviewer #1: I Don't Know

Reviewer #2: Yes

3. Have the authors made all data underlying the findings in their manuscript fully available?

Reviewer #1: Yes

Reviewer #2: Yes

4. Is the manuscript presented in an intelligible fashion and written in standard English?

Reviewer #1: Yes

Reviewer #2: Yes

5. Review Comments to the Author

Reviewer #1: This article continues a line of research that explores the interrelationship between various morphological traits in flake assemblages. The initial goal of this line of research was to find a set of variables that would determine the size of flake removals, thus allowing for the determination of the size of a flake before retouch to be determined. This study adds to the growing evidence that such a direct correlation is not possible as fracture involves multiple factors that interact in a complex fashion. In recent years studies of correlations of technological attributes are increasingly used as a means of developing and testing models of fracture propagation in brittle materials. This project presents an interesting new model that incorporates platform width as an important variable. My bias is that I would question the overall approach of using such correlations to investigate the process of brittle fracture which is a well developed branch of physics. But my qualms aside, this is a well-developed approach and the current article makes a significant contribution. In my recommendation for revisions I recommend to the authors to clarify the structure of their research more clearly. I would suggest that they first clearly delineate their model for fracture mechanics and then directly link this to the hypotheses that lead to the analysis. All of this material is currently in the paper but it is not clearly flagged for the reader to follow. The model of fracture dynamics are in a sense the most significant part of the paper so this should be highlighted. Similarly the hypotheses to be tested should be clearly delineated, and I was particularly confused on line 273 when we seemingly are told a new hypothesis that was going to be tested. Also, why do we wait to near the end of the paper to get a section on 'a new model'-- shouldn't this be up front as the model is not derived from the data but rather is the source of the hypotheses that are tested?

Methodologically the use of mediation analysis is key and I lack the expertise to evaluate the statistical analysis itself. However, there does need to be an explanation of why the authors do not use a statistical method used normally by physicists studying fracture mechanics. From the text it appears that this is a social sciences method and it is really unclear why it is being used in a materials science analysis. There needs to be a clear discussion of alternative statistical approaches available for study fracture mechanics and why mediation analysis was chosen as most appropriate.

I would add a few minor editorial points:

Line 67-69—seems a great overstatement to say that lithic manufacture has become synonymous with lithic technology and the role of replicative studies is exaggerated.

Line 77—not clear what platform setting refers to

Line 84-87—Again seems to be an exaggeration, makes it sound that shifts in EPA or PD control flake morphology independent of other factors. Thus in line 96-100 the authors contradict their earlier statement.

Line 143—Arrangement is the wrong word here

Reviewer #2: Lin et al. provide a well written and argued analysis of the role of platform width in the platform-flake relationship during knapping. Much attention has been devoted to platform attributes over the years, but platform width has only received passing mention. It is thus useful to see an exploration of the role platform width plays. The experimental methods and hypotheses were well chosen and explained, as was the role of mediator variables in understanding dependent/independent variables.

My only concern with the methods is a minor one, relating to the statistical tests chosen. The PW-PD ratios appear non-normally distributed (especially in fig8), for which parametric tests like ANOVA and Tukey are ill suited. It is likely that a Wilcoxon rank-sum test and post-hoc Kruskal-Wallis tests are more appropriate. I could be wrong, and maybe the data is parametric, but perhaps a brief mention that you tested the parametric criteria would be helpful.

While the implications of this study are discussed briefly in the intro and discussion, some more mention could be made of the broader significance of understanding platform-flake relationships. At the moment, it appears only relevant to others interested in fracture mechanics and the platform-flake relationship. For broader lithic/archaeological audiences it would be useful to explain the significance of understanding these relationships. What behavioral information does it provide? Can we draw conclusions about technological organization? Does the control of the flaking process by the knapper have cognitive/skill/behavioral implications etc.? A few sentences could help make this manuscript applicable to a broader audience.

L68: ‘very much’ can be deleted or replaced with something like ‘considerably’

L331: ‘casual’ should be ‘causal’, you may want to do a find and replace, Word may have autocorrected this a few times.

None of these comments detract substantively from the authors’ arguments and are only minor revisions. I highly recommend its publication in PLoS ONE. Thank you for the opportunity to review such an interesting article.

6. PLOS authors have the option to publish the peer review history of their article (what does this mean?). If published, this will include your full peer review and any attached files.

Reviewer #1: No

Reviewer #2: No

---

## [Author Response · Author response to Decision Letter 0]

29 Sep 2021

Editor comments:

- Please ensure the manuscript meets PLOS ONE style requirements.

 - Response: Removed numbering in headings within the manuscript. Corrected the naming of the figure files. Figures have been corrected using PACE.

- Please provide additional information regarding the specimens used in your study.

 - Response: No permits were required for this study. The relevant statement of declaration is outlined in lines 367-368. Information about the current locations of the studied experimental datasets have been added into the Material and Methods section (lines 228-230 and 320-321).

- Please ensure that you provide the correct grant numbers for the awards you received for your study in the ‘Funding Information’ section.

 - Response: ‘Funding Information’ section updated to match the information provided in ‘Financial Disclosure’.

- Please review your reference list to ensure that it is complete and correct.

 - Response: - All references reviewed. Minor corrections in error and formatting.

Reviewer #1 comments:

- My bias is that I would question the overall approach of using such correlations to investigate the process of brittle fracture which is a well developed branch of physics.

 - Response: We have included additional text in Introduction (lines 71-75) and Material and Methods (lines 253-259) to explain why we have taken a correlation approach here instead of using fracture mechanic models.

- I would suggest that they first clearly delineate their model for fracture mechanics and then directly link this to the hypotheses that lead to the analysis… The model of fracture dynamics are in a sense the most significant part of the paper so this should be highlighted.

 - Response: We have added text in Introduction (lines 179-184) to state our model of flake formation before outlining the specific hypotheses tested in this study.

- I was particularly confused on line 273 when we seemingly are told a new hypothesis that was going to be tested.

 - Response: This paragraph in Material and Methods has been removed.

- …why do we wait to near the end of the paper to get a section on 'a new model'-- shouldn't this be up front as the model is not derived from the data but rather is the source of the hypotheses that are tested?

 - Response: Text has been added in Introduction to explicitly state our hypothesised model up front (lines 179-184). The Discussion has been modified to make it clear that the section on the fracture model is an reiteration of our hypothesised model. The subheading of “A New Model of Fracture Initiation and Propagation” has been removed to avoid confusion.

- …there does need to be an explanation of why the authors do not use a statistical method used normally by physicists studying fracture mechanics. From the text it appears that this is a social sciences method and it is really unclear why it is being used in a materials science analysis… There needs to be a clear discussion of alternative statistical approaches available for study fracture mechanics and why mediation analysis was chosen as most appropriate.

 - Response: Text has been added to Material and Methods to explain why we employed a mediation analysis here (lines 259-262; 305-310). We have also included a new paragraph to provide more background information about mediation analysis, showing that the approach has been applied in fields beyond the social sciences.

- Line 67-69—seems a great overstatement to say that lithic manufacture has become synonymous with lithic technology and the role of replicative studies is exaggerated.

 - Response: This statement has been removed.

- Line 77—not clear what platform setting refers to.

 - Response: Reworded to ‘the shape of the platform surface.’

- Line 84-87—Again seems to be an exaggeration, makes it sound that shifts in EPA or PD control flake morphology independent of other factors. Thus in line 96-100 the authors contradict their earlier statement.

 - Response: We have modified the sentence in Line 86-87 so that it only summarises the experimental relationship.

- Line 143—Arrangement is the wrong word here.

 - Response: Reworded to ‘relationship.’

Reviewer #2 comments

- The PW-PD ratios appear non-normally distributed (especially in fig8), for which parametric tests like ANOVA and Tukey are ill suited. It is likely that a Wilcoxon rank-sum test and post-hoc Kruskal-Wallis tests are more appropriate.

 - Response: We have specified in our original text that all variables were transformed to achieve an approximately symmetrical distribution to fulfill the assumption of the statistical tests used (e.g., lines 377-378 and the caption for Fig 8). 

- …some more mention could be made of the broader significance of understanding platform-flake relationships.

 - Response: Additional text has been added in Discussions (lines 502-506) to relate the study results to broader topics of lithic technology.

- L68: ‘very much’ can be deleted or replaced with something like ‘considerably’.

 - Response: Removed.

- L331: ‘casual’ should be ‘causal’, you may want to do a find and replace, Word may have autocorrected this a few times.

 - Response: Corrected throughout the text.

---

## [Decision Letter · Decision Letter 1]

9 Nov 2021

PONE-D-21-21600R1The mediating effect of platform width on the size and shape of stone flakesPLOS ONE

Dear Dr. Lin,

Thank you for submitting your manuscript to PLOS ONE. After careful consideration, we feel that it has merit but does not fully meet PLOS ONE’s publication criteria as it currently stands. Therefore, we invite you to submit a revised version of the manuscript that addresses the points raised during the review process.

We look forward to receiving your revised manuscript.

Kind regards,

Peter F. Biehl, PhD

Academic Editor

PLOS ONE

Journal Requirements:

Additional Editor Comments:

All comments raised by reviewer 1 need to be addressed before the manuscript can be accepted.

Reviewers' comments:

Reviewer's Responses to Questions

**Comments to the Author**

1. If the authors have adequately addressed your comments raised in a previous round of review and you feel that this manuscript is now acceptable for publication, you may indicate that here to bypass the “Comments to the Author” section, enter your conflict of interest statement in the “Confidential to Editor” section, and submit your "Accept" recommendation.

Reviewer #1: (No Response)

Reviewer #2: All comments have been addressed

2. Is the manuscript technically sound, and do the data support the conclusions?

Reviewer #1: Partly

Reviewer #2: Yes

3. Has the statistical analysis been performed appropriately and rigorously? 

Reviewer #1: I Don't Know

Reviewer #2: Yes

4. Have the authors made all data underlying the findings in their manuscript fully available?

Reviewer #1: Yes

Reviewer #2: Yes

5. Is the manuscript presented in an intelligible fashion and written in standard English?

Reviewer #1: Yes

Reviewer #2: Yes

6. Review Comments to the Author

Reviewer #1: The authors have made minimal editorial changes to the text but have done little to address fundamental concerns with this article. I should note up front that I am not familiar with the specific statistical method used here and I found the authors’ explanation difficult to follow. This limitation should be kept in mind when assessing my review.

The fundamental issue with this paper is that it claims to be finding causal relationships between variables based on correlations. Thus the authors argue that they are making fundamental observations about fracture mechanics while not relying on any methods from the study of fracture mechanics. In fact they seem to imply that understanding brittle fracture is hopeless so this is the best we can do. The logic here seems muddy to me and I urge the authors to give this further thought. Towards that end it important to keep in mind that the goal of most studies of platform attributes is not to create model of fracture mechanics but rather to find correlations that would allow for the prediction of the morphology of incomplete flakes.

My sense is that the authors’ have not quite identified what it is that they are actually doing in their study. They have definitively not determined a chain of causal factors controlling fracture in brittle solids. Such models are well published in the scientific literature and they do not look anything like what is presented here. If I hazard a guess, I think what interests these authors, and what interests me as well, is not the actual physics of fracture mechanics but rather the visually accessible attributes that would be available to a knapper attempting to control the attributes of knapped flakes. Thus their work is focused not on the fracture mechanics themselves (a process that takes place at the atomic and molecular scale) but rather the cues that would be available to a knapper in the process of flake production. In that framework the data presented here makes sense as the causal chain would relate to the knapper’s inference of a causal chain of related attributes.

I hope these comments are constructive and that the authors will consider thinking about the underlying structure of their inquiry. My sense is that they are trying to get at something, and have produced relevant data, that is important but that they have not clarified what their goals actually are.

Reviewer #2: (No Response)

7. PLOS authors have the option to publish the peer review history of their article (what does this mean?). If published, this will include your full peer review and any attached files.

Reviewer #1: No

Reviewer #2: No

---

## [Author Response · Author response to Decision Letter 1]

10 Dec 2021

Reviewer 1 comment: 

The fundamental issue with this paper is that it claims to be finding causal relationships between variables based on correlations. Thus the authors argue that they are making fundamental observations about fracture mechanics while not relying on any methods from the study of fracture mechanics. 

My sense is that the authors’ have not quite identified what it is that they are actually doing in their study. They have definitively not determined a chain of causal factors controlling fracture in brittle solids.

I think what interests these authors, and what interests me as well, is not the actual physics of fracture mechanics but rather the visually accessible attributes that would be available to a knapper attempting to control the attributes of knapped flakes. Thus their work is focused not on the fracture mechanics themselves (a process that takes place at the atomic and molecular scale) but rather the cues that would be available to a knapper in the process of flake production. In that framework the data presented here makes sense as the causal chain would relate to the knapper’s inference of a causal chain of related attributes.

My sense is that they are trying to get at something, and have produced relevant data, that is important but that they have not clarified what their goals actually are.

Response: 

We agree with the reviewer that our study is concerned about the relationship between knapping factors and flaking outcome, rather than the physical processes of fracture mechanics themselves. We have included additional text in the manuscript to make this distinction clear. For example, after outlining our proposed conceptual model of platform formation in Introduction, we added the following text (lines 191-194): 

“Undoubtedly, this model is simplistic and does not represent the actual mechanisms underlying the fracture process. However, as a conceptual heuristic, the model can help us hypothesize about the interrelationship between platform variables, such as PD and PW, and flake attributes.”

A similar section is now included Material & Methods (lines 265-273): 

“We acknowledge that this approach does not investigate the actual causal mechanisms underlying the fracture process – such mechanisms will need to be understood from principles of fracture mechanics. Instead, we are interested in characterizing the relationships among the flake variables in question here through statistical modeling... Note that terms such as ‘effect’, ‘influence’ and ‘impact’ are used here strictly to describe statistical relationships among the independent and dependent test variables, rather than the underlying causal mechanisms of fracture.”

We have also made changes throughout the manuscript to make clear the distinction thtat we are mainly discussing statistical relationships rather than the causal mechanisms of flake fracture. These changes mainly occurred in Results and Discussion.

---

## [Editor Report · Decision Letter 2]

10 Jan 2022

The mediating effect of platform width on the size and shape of stone flakes

PONE-D-21-21600R2

Dear Dr. Lin,

We’re pleased to inform you that your manuscript has been judged scientifically suitable for publication and will be formally accepted for publication once it meets all outstanding technical requirements.

Kind regards,

Peter F. Biehl, PhD

Academic Editor

PLOS ONE
---

## [Editor Report · Acceptance letter]

12 Jan 2022

PONE-D-21-21600R2 

The mediating effect of platform width on the size and shape of stone flakes 

Dear Dr. Lin:

I'm pleased to inform you that your manuscript has been deemed suitable for publication in PLOS ONE. Congratulations! Your manuscript is now with our production department. 

Kind regards, 

on behalf of

Dr. Peter F. Biehl 

Academic Editor

PLOS ONE